# Lactoferrin and Lysozyme Inhibit the Proteolytic Activity and Cytopathic Effect of *Naegleria fowleri* Enzymes

**DOI:** 10.3390/pathogens13010044

**Published:** 2024-01-03

**Authors:** Moises Martinez-Castillo, Gerardo Ramírez-Rico, Mineko Shibayama, Mireya de la Garza, Jesús Serrano-Luna

**Affiliations:** 1Liver, Pancreas and Motility Laboratory, Unit of Research in Experimental Medicine, School of Medicine, Autonomous National University of Mexico (UNAM), Mexico City 06720, Mexico; mpcastillo@comunidad.unam.mx; 2Department of Cell Biology, Center for Research and Advanced Studies, Mexico City 07360, Mexico; garmvz@gmail.com (G.R.-R.); mireya.dela.garza@cinvestav.mx (M.d.l.G.); 3Faculty of Professional Studies Cuautitlan, Autonomous National University of Mexico, Mexico City 54714, Mexico; 4Department of Infectomics and Molecular Pathogenesis, Center for Research and Advanced Studies, Mexico City 07360, Mexico

**Keywords:** *Naegleria fowleri*, lactoferrin, lysozyme, amoebicidal effect, protease activity

## Abstract

*Naegleria fowleri* is a ubiquitous free-living amoeba that causes primary amoebic meningoencephalitis. As a part of the innate immune response at the mucosal level, the proteins lactoferrin (Lf) and lysozyme (Lz) are secreted and eliminate various microorganisms. We demonstrate that *N. fowleri* survives the individual and combined effects of bovine milk Lf (bLf) and chicken egg Lz (cLz). Moreover, amoebic proliferation was not altered, even at 24 h of co-incubation with each protein. Trophozoites’ ultrastructure was evaluated using transmission electron microscopy, and these proteins did not significantly alter their organelles and cytoplasmic membranes. Protease analysis using gelatin-zymograms showed that secreted proteases of *N. fowleri* were differentially modulated by bLf and cLz at 3, 6, 12, and 24 h. The bLf and cLz combination resulted in the inhibition of *N. fowleri*-secreted proteases. Additionally, the use of protease inhibitors on bLf-zymograms demonstrated that secreted cysteine proteases participate in the degradation of bLf. Nevertheless, the co-incubation of trophozoites with bLf and/or cLz reduced the cytopathic effect on the MDCK cell line. Our study suggests that bLf and cLz, alone or together, inhibited secreted proteases and reduced the cytopathic effect produced by *N. fowleri;* however, they do not affect the viability and proliferation of the trophozoites.

## 1. Introduction

*Naegleria fowleri* is the etiological agent of primary amoebic meningoencephalitis (PAM), with a 95% death rate, mainly in children and young adults with a recent history of swimming. Amoeba access to the host occurs via the nasal route, and the trophozoites travel through the olfactory neuroepithelium and cross the cribriform plate until they reach the olfactory bulbs, causing intense tissue damage [1]. The nasal mucosa filters approximately seven liters of air per minute and is a barrier against foreign particles and microorganisms [2]. The elimination of potential pathogens is related to the presence of antimicrobial proteins, including mucins, antimicrobial peptides, lysozymes, lactoferrins, and immunoglobulins [3]. However, *N. fowleri* uses several pathogenic mechanisms to evade the innate barrier [4,5,6]. Additionally, the risk of PAM infection has been calculated in different regions worldwide, suggesting that the number of trophozoites may be correlated with the acquisition of infection. However, not all individuals exposed to contaminated water acquire this parasite [7,8,9]. Genetic and host defense mechanisms may play an important role in establishing the infection (e.g., poor innate immune response, rhinitis, allergies, and alterations or deficiencies in immune responses) [1,10]. 

Lactoferrin (Lf) is an iron-binding glycoprotein of the mammalian innate immune system [11]. This protein can be found bound to iron (holo-lactoferrin) and unbound (apo-lactoferrin, apo-Lf). Lf regulates iron homeostasis, controls microbial infections, exhibits anti-inflammatory activity, and protects against cancer [12,13]. Lf has direct and indirect microbicidal effects. The direct association of apo-Lf with lipopolysaccharides, porins, and teichoic acids alters the permeability of Gram-negative and Gram-positive bacteria [14,15,16]. Additionally, it has been reported that the chelation of iron by apo-Lf reduces the proliferation of yeasts, fungi, viruses, and protozoa [12,17]. Lysozyme (Lz) was reported by Alexander Fleming as the main protein with intrinsic antimicrobial properties of human nasal secretions [18]. In general, Lz can induce the hydrolysis of 1,4-beta-linkages between N-acetylmuramic acid and N-acetyl-D-glucosamine residues in bacterial peptidoglycans [19]. It has also been reported that patients with perennial allergic rhinitis and recurrent sinusitis display lower Lz concentrations than healthy individuals [20].

The combined antimicrobial effects of Lf and Lz have only been reported for some protozoan parasites. In this context, it was demonstrated in *Entamoeba histolytica* that the individual and combined effects of human apo-Lf, Lz, and IgA (1 mg/mL of each protein) promoted a reduction in amoebic viability in vitro after 3 h of co-incubation [21]. In *Giardia intestinalis*, it was reported that bovine Lf (bLf) and their N-terminal peptides produced giardicidal activity. The authors demonstrated that approximately 15 µM (1.2 mg/mL) of bLf is a sufficient concentration to decrease the viability of trophozoites by 80% [22]. In contrast, when the intracellular parasitic protozoan *Toxoplasma gondii* was pre-incubated with bLf (1 mg/mL), the parasites penetrated mouse peritoneal macrophages and mouse embryonic cells, demonstrating that Lf did not have a parasiticidal effect on *T. gondii*. However, when embryonic cells and macrophages were treated with apo-bLf or holo-bLf after inoculation with *T. gondii*, inhibition of the intracellular development of parasites was observed [23]. 

Regarding *N. fowleri,* it has been reported that apo-bLf (1 mg/mL) does not induce apoptosis after 12 h of incubation. However, the effects of physiological and higher concentrations of this protein have not been explored [24]. The present study focused on the individual and combined effects of Lf and chicken egg Lz (cLz) on *N. fowleri* trophozoites. Trophozoites resisted the amoebicidal and amoebostatic effects of both antimicrobial proteins under physiological conditions, even at higher concentrations. Moreover, cysteine proteases were associated with amoeba survival. However, co-incubation of trophozoites with these antimicrobial proteins significantly reduced the cytopathic effect of *N. fowleri* on an epithelial monolayer (MDCK cells). The correct function of the host immune response plays an important role in the progression or neutralization of microbial virulence factors, avoiding the invasion of the target cells; however, the broad spectrum of pathogenic mechanisms of *N. fowleri* also plays a crucial role in the survival, migration, and invasion of the central nervous system.

## 2. Materials and Methods

### 2.1. Cell Cultures

The pathogenic KUL strain *N. fowleri* (ATCC 30808), isolated from cerebrospinal fluid, was used in all the experiments. The culture was maintained in axenic conditions in 2% (*w*/*v*) Bactocasitone medium (BD Gibco, Waltham, MA, USA), supplemented with 10% (*v*/*v*) fetal bovine serum (FBS; Equitech-bio, Kerrville, TX, USA) at 37 °C. The trophozoites were harvested after 48 h during the logarithmic growth phase. MDCK cells were maintained in minimal essential medium (MEM; Gibco Life Technologies, Waltham, MA, USA) with 10% (*v*/*v*) FBS (Hyclone, Washington, DC, USA) in a 5% CO_2_ atmosphere at 37 °C.

### 2.2. Viability Assays 

Amoebic cultures were incubated with bLf (5 mg of iron/100 g, NutriScience, Milford, CT, USA), cLz (Sigma-Aldrich, St. Louis, MI, USA), or both proteins, and cell viability was determined using the trypan blue assay. Briefly, 1 × 10^6^ trophozoites were incubated with different concentrations of bLf, diluted in Bactocasitone medium (5, 10, 25, 50, 100, 300, and 500 µM) and with cLz (10, 15, and 20 µM) during 0, 3, 6, 12, 18, and 24 h. bLf and cLz were diluted in amoebic culture medium without FBS. For the trypan blue assay, and the trophozoites were harvested after each interaction. Then, the amoeba was incubated with 0.5% trypan blue (Sigma-Aldrich, St. Louis, MI, USA); at least 100 cells from different microscopic fields were analyzed using light microscopy to evaluate viability. Data are reported as the percentage of live amoebae. *N. fowleri* free-FBS culture was used as the experimental control. Three independent cell viability assays were performed. Statistical analysis was performed using one-way analysis of variance (ANOVA) using GraphPad Prism 5.0 (GraphPad, San Diego, CA, USA).

### 2.3. Growth Curves

Cell proliferation and mitochondria metabolic activity were evaluated using a 3-(4,5-dimethylthiazol-2-yl)-2,5-diphenyltetrazolium bromide (MTT) assay (Sigma-Aldrich, St. Louis, MI, USA). Trophozoites were incubated or not with the antimicrobial proteins at a maximum concentration of 500 µM (bLf) and 20 µM (cLz) at 0, 3, 6, 12, 18, and 24 h; trophozoites were also incubated in medium without cLz or bLf. The optical density was determined using an Epoch microplate reader spectrophotometer (BioTek, Winooski, VT, USA) at 570 nm. Data were compared with an MTT standard curve of previously obtained trophozoites (5 × 10^4^–2.5 × 10^6^). Two-way ANOVA was performed using Systat Sigma Plot v15 software (SYSTAT, San Jose, CA, USA). Graphs were generated using GraphPad Prism 5 software. The data are expressed as the mean ± standard deviation (SD) of three independent experiments.

### 2.4. Transmission Electron Microscopy

Transmission electron microscopy (TEM) was used for ultrastructural analysis of the morphological changes induced by bLf and cLz and their possible synergy with trophozoites. Trophozoites were incubated with the combination of bLf (500 µM) and cLz (20 µM) for 12, 18, and 24 h and fixed with 2.5% glutaraldehyde (Sigma-Aldrich, St. Louis, MI, USA) in 0.1 M sodium cacodylate buffer (pH 7.2) (Sigma-Aldrich, St. Louis, MI, USA). Samples were post-fixed with 2% osmium tetroxide. They were then dehydrated with increasing ethanol concentrations and embedded in epoxy resin (Sigma-Aldrich, St. Louis, MI, USA). Semi-thin sections were stained with 0.5% toluidine blue. The thin sections were stained with uranyl acetate and lead citrate (Electron Microscopy Sciences, Hatfield, PA, USA) and analyzed under a microscope (Jeol JEM-1400; TYO, Jeol, AKISHIMA, Japan).

### 2.5. Zymography Assays

Protease activity was examined in 10% sodium dodecyl sulfate-polyacrylamide gels copolymerized with porcine gelatin or bLf. The final substrate concentration was 0.1% (*w*/*v*). Total crude extracts (TCEs) of trophozoites were incubated with or without bLf (500 µM), with cLz (20 µM), or with the combination of bLf (500 µM) plus cLz (20 µM) over 1, 3, 6, and 12 h; these extracts were obtained as previously described [25]. Briefly, trophozoites were removed from the culture flask surface by chilling in an ice bath for 30 min, centrifuging at 800× *g* for 10 min, and washing with phosphate-buffered saline (PBS; pH 7.2). Subsequently, the pellets were incubated at 37 °C for 30 min and lysed in PBS (pH 7.2) through six freeze–thaw cycles with hot water and liquid nitrogen. The conditioned medium (CM) was prepared according to the following protocol. Six million trophozoites were placed into culture flasks that contained 3 mL of fresh free-FBS Bactocasitone, and then the amoebae were incubated at 37 °C for 12 h. The CM was obtained through centrifugation at 1500× *g* for 10 min and finally passed through a 0.22 µm Durapore membrane (Millipore, Burlington, MA, USA). Subsequently, samples were precipitated with absolute ethanol at a 3:1 ratio and stored at −20 °C for 2 h. The CM was centrifuged at 6500× *g* for 45 min. Protein concentrations in TCE and CM were quantified using the Bradford method [26]. Forty micrograms of protein was loaded in the gel, and the electrophoresis was performed at 4 °C in an ice bath at 80 V for 1 h; gels were washed twice with 2.5% (*v*/*v*) Triton X-100 solution (Sigma-Aldrich) for 30 min. The gels were incubated overnight in 100 mM Tris-HCl (pH 7.0). Finally, gels were stained overnight with 0.5% (*w*/*v*) Coomassie Brilliant Blue R-250 (Sigma-Aldrich). The protease activity was identified as a clear band on a blue background.

### 2.6. Protease Inhibition

For inhibition assays, TCE and CM proteins from trophozoites were incubated for 1 h at 37 °C with or without protease inhibitors in constant agitation previous to zymogram assays using 0.1% (*w*/*v*) bLf as substrate. The concentrations of the inhibitors were as follows: for cysteine proteases, 10 mM p-hydroxymercuribenzoate (pHMB, Sigma-Aldrich, St. Louis, MI, USA) and 10 µM trans-epoxysuccinyl-L-leucylamido [4-guanidino] butane (E-64) (Roche, Basel, Switzerland); for serine proteases, 5 mM phenyl-methyl sulfonyl fluoride (PMSF, Sigma-Aldrich, St. Louis, MI, USA) and 1 mM aprotinin (Sigma-Aldrich, St. Louis, MI, USA).

### 2.7. Cytopathic Effect of N. fowleri on MDCK Cells

The cytopathic effect of *N. fowleri* was determined in a monolayer of MDCK epithelial cells, with or without bLf (500 µM), cLz (20 µM), or the combination of both proteins during 12 h of incubation. Confluent cultures of MDCK cells (98%) were maintained in MEM (Gibco Life Technologies, Waltham, MA, USA), and the medium was replaced with serum-free MEM. Co-incubation with *N. fowleri* trophozoites (1 × 10^6^) was performed in 24-well culture plates (1:1) (Corning) in the presence of bLf, cLz, or both. After 12 h, trophozoites were removed by chilling the culture plates for 30 min and washing with PBS, followed by the addition of methylene blue dye 1% (*v*/*v*) (Sigma-Aldrich, St. Louis, MI, USA). The supernatants were recovered, and the absorbance was determined using a microplate reader spectrophotometer at 660 nm. MDCK cells without trophozoites, MDCK cells treated with antimicrobial proteins, and MDCK cells incubated with trophozoites were used as controls. Statistical analysis (two-way ANOVA) was performed using the Systat Sigma Plot software. Graphs were generated using GraphPad Prism 5 software. The data are presented as the mean ± SD of three independent experiments.

## 3. Results

### 3.1. bLf and cLz Did Not Have an Amoebicidal Effect over N. fowleri Trophozoites

The viability of *N. fowleri* was evaluated in the presence of bLf, cLz, and the combination of both antimicrobial proteins. Trypan blue assays showed that the viability of trophozoites was approximately 95% at 0, 3, 6, 12, 18, and 24 h with 500 µM of bLf and 20 µM of cLz either alone or combined. Similar results were observed with 5, 10, 25, 50, 100, and 300 µM of bLf and 10 and 15 µM of cLz (Appendix A). The viability of untreated trophozoites in the serum-free medium was estimated to be 98%, even after 24 h of culture (Figure 1a). Three independent assays were performed with similar results; bars display the mean ± SD of three independent assays (*p* > 0.05). 

### 3.2. bLf and cLz Did Not Affect the Growth of N. fowleri Trophozoites 

It has been reported that the evaluated antimicrobial proteins can induce a temporal reduction in the proliferation of bacteria when a requirement is absent (bacteriostatic effect). Therefore, we decided to evaluate the capacity of antimicrobial proteins in amoebic proliferation. Our results revealed that *N. fowleri* trophozoites incubated with a high concentration of bLf and/or cLz did not alter their proliferation compared with the control cultures at any time (Figure 1b–d). The differences were not statistically significant under any condition (*p* > 0.05). 

### 3.3. Ultrastructural Analysis of Trophozoites 

To evaluate membrane changes in *N. fowleri* trophozoites, we conducted an ultrastructural analysis of *N. fowleri* trophozoites incubated with a combination of bLf and cLz for 12, 18, and 24 h. TEM analysis showed that untreated trophozoites presented typical ultrastructural characteristics of the nucleus and nucleolus, with several mitochondria and vacuoles in their cytoplasm, and that the cytoplasmic membrane was intact at 12 h (Figure 2a). In contrast, *N. fowleri* trophozoites treated with a combination of bLf and cLz showed evident morphological changes at 12 h, including the loss of their form and the presence of numerous vacuoles with cytoplasmic contents (arrows), similar to autophagy vacuoles (Figure 2b). Additionally, some trophozoites displayed high density on the cytoplasmic membrane. These changes were observed in approximately 60% of trophozoites; however, no damage or evidence of lysis was observed. Furthermore, at 18 h, the trophozoites recovered their morphology and vacuoles, with a decrease in their previous content (arrows), and less than 10% of *N. fowleri* trophozoites displayed membrane discontinuity (Figure 2c). After 24 h of incubation, the combination of antimicrobial proteins seemed to stop affecting the trophozoites, which showed a gradual recovery of their shape with an evident clearance of the vacuolar content, the presence of a normal nucleus and nucleolus, and several apparently normal mitochondria (Figure 2d). Individual co-incubation showed similar characteristics to the amoebic culture.

### 3.4. Regulation of Amoebic Proteases by the Antimicrobial Proteins

Zymograms of TCE and CM from amoebic cultures co-incubated with bLf and/or cLz were obtained. Proteolytic patterns in gelatin (a general substrate) copolymerized gels. TCE from *N. fowleri* incubated in a serum-free medium showed at least three protease activities of 150, 100, and 75 kDa (Figure 3a, lane 1). TCE from *N. fowleri* treated with bLf (500 µM) did not display differences in protease activities regarding the control at 3, 6, and 12 h (Figure 3a, lanes 2–4). In contrast, evident differences were observed in the zymogram of the CM. Control activities of CM from amoebic cultures without bLf showed clear bands at 170, 130, 80, and 60 kDa (Figure 3b, lane 1), whereas CM from trophozoites treated with bLf showed bands at 170 and 60 kDa at 3 h (Figure 3b, lane 2). Moreover, the downregulation of proteolytic bands at 170, 130, and 80 kDa was observed at 6 and 12 h (Figure 3b, lanes 3 and 4). 

Zymograms of TCE from trophozoites incubated without cLz showed main activities of 150, 100, and 75 kDa (Figure 4a, lane 1). Moreover, TCE incubated with cLz (20 µM) showed a significant reduction in all proteolytic activities at 3 and 6 h (Figure 4a, lanes 2 and 3), and later, at 12 h, proteolytic activities of 100 and 75 kDa reappeared (Figure 4b, lanes 3 and 4). In the CM of trophozoites incubated without cLz, only bands of 150 and 100 kDa were appreciated (Figure 4b, lane 1); when CM was incubated with cLz (20 µM), proteolytic bands of 100 and 75 kDa appeared, increasing the activities with time (Figure 4b, lanes 2–4). The evaluation of the proteolytic pattern of the combination of bLf (500 µM) and cLz (20 µM) incubated with TCE of *N. fowleri* showed a weak activity of 170 kDa (Figure 5b, lanes 2–4), whereas in TCE incubated with both proteins, a weak activity of 150 kDa was observed in all assayed times (Figure 5a, lanes 2–4). 

### 3.5. Cysteine Proteases from N. fowleri Degrade Bovine Lactoferrin 

To evaluate the degradation of specific substrates, zymogram assays were performed using bLf. No proteolytic activity was observed in TCE (Figure 6a, lane 1). Proteolytic activity was observed in the CM from trophozoites treated for 24 h with bLf. The zymograms of bLf revealed bands at 150 and 73 kDa (Figure 6a, lane 2). Additionally, we determined that cysteine proteases were involved in the degradation of bLf. Specific inhibitors of cysteine proteases (E-64 and pHMB) abrogated these activities (Figure 6b, lanes 1 and 2); however, proteolytic activity was still observed with serine protease inhibitors (PMSF and aprotinin) (Figure 6b, lanes 3 and 4). Zymograms of cLz were not obtained, because copolymerization of acrylamide with this antimicrobial protein was not successful. 

### 3.6. Inhibition of Cytopathic Effect of N. fowleri by bLf and cLz 

To determine whether *N. fowleri* trophozoites incubated with bLf and cLz could damage the MDCK cell monolayer, we evaluated the cytopathic effect using methylene blue staining, which was quantified spectrophotometrically. The higher the viability/confluence of MDCK, the lower the cytopathic effect of *N. fowleri* proteases (and the higher the inhibitory effect of peptides). Our results showed that the incubation of *N. fowleri* trophozoites directly onto the MDCK cell monolayer led to substantial host cell death, with an over 80% decrease in epithelial cell viability after 12 h of co-incubation (Figure 7). The co-incubation with bLf (500 µM), cLz (20 µM), or with both proteins reduced MDCK viability by approximately 50%, while dual co-incubation with bLf and cLz avoided cell death by up to 80%. We used MDCK cells without amoebae and with the antimicrobial proteins as controls. The results showed that the monolayers maintained their integrity under control and experimental conditions with 98% viability. Based on these results, we concluded that bLf and cLz inhibit the cytopathic effects of live trophozoites in the epithelial monolayer.

## 4. Discussion

In the nasal mucosa, several immune proteins participate actively in the elimination of pathogens, including mucins, macroglobulins, immunoglobulins (mainly IgA), antimicrobial proteins, and peptides, such as lysozyme, lactoferrin, and defensins [3]. Iron-free lactoferrin is the form released by mucoepithelial cells and neutrophils [11,27]. This protein is implicated in several physiological roles, such as the regulation of iron levels, anti-inflammatory activity, regulation of cellular growth, cell differentiation, protection against cancer, and host defense against a broad range of microbial infections [13,27]. Additionally, in the nasal mucus, one of the most important antimicrobial proteins is lysozyme, which has been shown to, alone or in combination with innate, intrinsic proteins (antimicrobial proteins: lactoferrin, cathelicidin, and defensins) and exogen compounds (e.g., flavonoids and imidazole), display a broad antimicrobial and micro-biostatic activity against bacteria and fungi [28,29]. 

During experimental PAM, trophozoites are embedded in the mucus and surrounded by neutrophils during the initial stages of infection [6,30]. Therefore, in the present study, the antimicrobial properties of bLf and cLz, alone and in combination, were evaluated in *N. fowleri* cultures. Previously, it was reported that bLf (1.25 µM) does not cause significant negative effects at 12 h of interaction [24]. However, a detailed study on bLf at physiological and high concentrations has not yet been performed. 

Lf concentration in human colostrum has been estimated at 15 mg/mL (187 µM) and in human milk at approximately 1 mg/mL (12.5 µM) [22]. Moreover, the gallbladder can release approximately 10 to 30 mg/mL (125–375 µM) into the duodenal lumen. In nasal fluids, the normal concentration of apo-bLf is estimated at 1 to 2.5 µM and 15 to 35 µM for Lz; however, this concentration increases during microbial infection [20]. The microbicide effect of these proteins has been shown in *E. histolytica*, where the authors reported that apo-bLf concentrations of 12.5 µM diminish to 40% the viability of trophozoites after 3 h of co-incubation and 70 µM of cLz reduces to 20%, while the synergy of these antimicrobial proteins plus IgA (1 mg/mL) reduces the viability to 10% [31]. We found that the viability of *N. fowleri* was not affected by bLf or cLz at physiological or higher concentrations. It is important to mention that the solution of cLz at a higher concentration of 20 µM forms crystal particles, so we only used the physiological concentration of cLz. *Acanthamoeba castellanii*, another free-living amoeba, also showed negative effects of human milk lactoferrin (10 µg) and Lz (2.5 µg) [32]. Moreover, Lf and Lz alone can induce a microbiological effect associated with free iron chelation and changes in pH [15]. The inhibition of proliferation has been well reported in bacteria, fungi (*Candida* spp.), and protozoans, including *T. gondii*, *Trypanosoma brucei*, *T. cruzi*, and *G. lamblia* [33,34,35], but the molecular and cellular mechanisms are not fully understood. In our study, *N. fowleri* growth curves did not show alterations in proliferation under any culture conditions. To determine whether the capping and shedding of *N. fowleri* participate in resistance to amoebicidal and amebostatic effects, we performed ultrastructural analysis of trophozoites incubated with each antimicrobial protein. Both bLf and Lz have cationic characteristics that allow them to bind to cationic sites on microbial membranes [14,27,36]. Anionic sites in *Naegleria gruberi* (a nonpathogenic species) have been identified using ferritin as a cationic protein [37]. Herein, we found an atypical rearrangement of the membrane and the presence of continuous deposition of an electron-dense material during the interaction with bLf; however, specific structures suggesting capping or damage were not observed. Furthermore, cytoplasmic alterations or damage to organelles (mitochondria, nuclei, and nucleoli) have not yet been determined [38].

After observing the resistance of trophozoites to bLf and/or cLz, we evaluated how trophozoites avoid the microbicidal effects of these innate immune molecules. To accomplish this, we analyzed the role of amoebic proteases, an extensively studied pathogenic mechanism [39,40,41]. The inactivation of bLf and cLz by proteolytic activity has been reported in oral tract bacteria associated with successful colonization [42,43,44]. To determine whether proteases from *N. fowleri* participate in bLf and cLz resistance, we evaluated the proteolytic patterns under different conditions using zymography. Zymogram assays indicated the differential modulation of proteases present in TCE and CM. Moreover, zymograms employing gelatin as a substrate (general protease substrate) and bLf (specific substrate) showed that bLf induced the secretion of specific proteases [45,46,47]. In *N. fowleri* trophozoites, the production and secretion of proteases mainly involve cysteine and serine proteases [25,48,49,50]. Herein, protease characterization was correlated with the activity of cysteine proteases in bLf. Similar results were reported in gelatin-zymograms of cysteine proteases from *E. histolytica*, which can cleave holo-Lf at pH 7.0 and 4.0 [51]. 

Finally, we evaluated whether the regulation of proteases was associated with the pathogenicity of trophozoites and their cytopathic effects on epithelial cells [50]. The cytopathic effects of *N. fowleri* proteases on epithelial cells were reduced in the presence of bLf and cLz. Similar results were recently reported by our group in *A. castellanii*, where the co-incubation of *A castellani* trophozoites with bLf prevented damage to MDCK cells. Moreover, we conclude that proteases are differentially regulated in the presence of bLf [52]. Several reports provide evidence that proteases are important virulence factors in free-living pathogens [53]. Probably, bLf and cLz can bind directly to proteases, consequently reducing their activities. Thus, these proteins may provide partial protection during PAM infection, reducing tissue damage, whereas inefficient activation of these molecules and other mediators of the innate immune response increases susceptibility to infection. Future studies should evaluate the roles of bLf and cLz in the nasal cavity using in vivo models.

Our results demonstrate that *N. fowleri* is one of the few microorganisms that resist the effects of bLf and cLz, alone or in combination with the secretion of cysteine proteases. These antimicrobial proteins can significantly reduce the cytopathic effect of *N. fowleri* trophozoites; therefore, the pathogenic mechanism of the amoeba and the immune response triggered by the host play important roles in the survival of the amoeba, damage to host tissues, and establishment of infection. 

## 5. Conclusions

Our study strongly suggests that bLf and cLz can inhibit secreted *N. fowleri* proteases, causing a decrease in epithelial cell damage, but are inefficient in affecting the viability and proliferation of trophozoites, allowing for a decrease in epithelial cell damage. 

## Figures and Tables

**Figure 1 pathogens-13-00044-f001:**
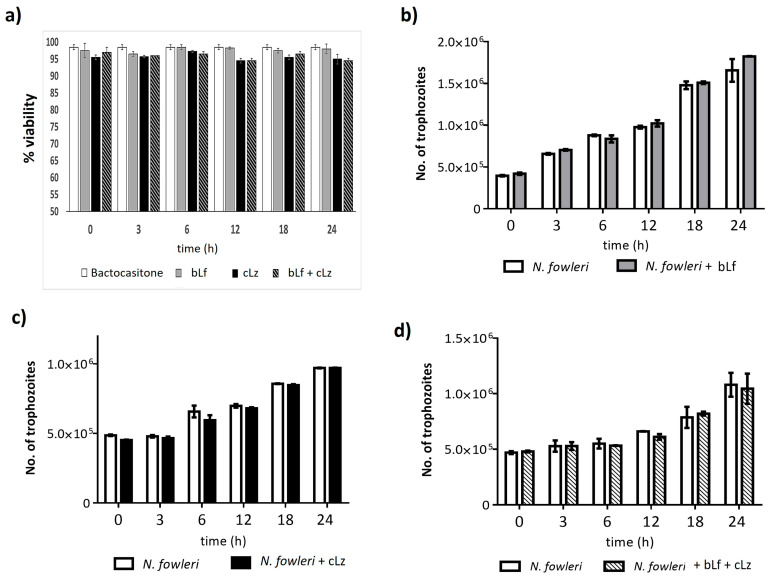
Viability and growth curves of *N. fowleri* trophozoites incubated with antimicrobial proteins. (**a**) Viability of trophozoites using Trypan blue (0.2%) assays of amoebic cultures co-incubated with bLf (500 µM), cLz (20 µM), and combination of both at 0, 3, 6, 12, 18, and 24 h. As an experimental control, amoebic cultures in Bactocasitone without FBS were used (98% viability). MTT proliferation curves of trophozoites incubated with (**b**) bLf (500 µM), (**c**) cLz (20 µM), and (**d**) combined at different incubation times. Bactocasitone medium without FBS was used as an experimental control. The data analyses were performed with GraphPad Prism 5.0. Bars show the mean ± SD of three independent assays (*p* > 0.05).

**Figure 2 pathogens-13-00044-f002:**
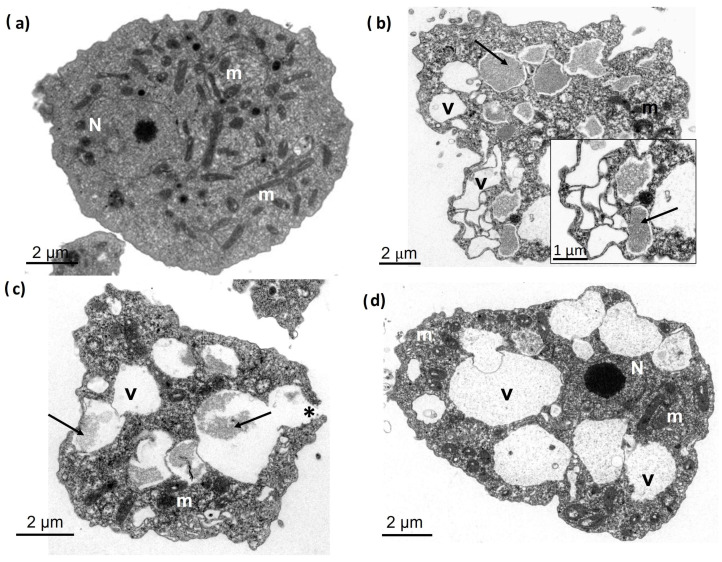
Transmission electron microscopy of *N. fowleri* after interaction with antimicrobial proteins. (**a**) *N. fowleri* trophozoite incubated in free-FBS medium at 12 h shows the integrity of organelles and cytoplasmic membrane. (**b**) *N. fowleri* treated with the combination of bLf (500 µM) and cLz (20 µM) at 12 h presents morphological changes, vacuoles (V) with cytoplasmatic content (arrows), and normal mitochondria and continuous membrane are observed. (**c**) Trophozoite after 18 h of co-incubation with both bLf and cLz showed vacuoles (V) with fibrillar content (arrows), a smaller number of mitochondria without apparent alterations, cytoplasmic membrane presents a discontinuity (asterisk, less than 10% of analyzed amoebas). (**d**) At 24 h of co-incubation with both proteins, it was observed a normal nucleus (N) with prominent nucleolus, the integrity of the membrane, multiple normal mitochondria, and several empty vacuoles (V) present in the cytoplasm. Bars = 2 µm.

**Figure 3 pathogens-13-00044-f003:**
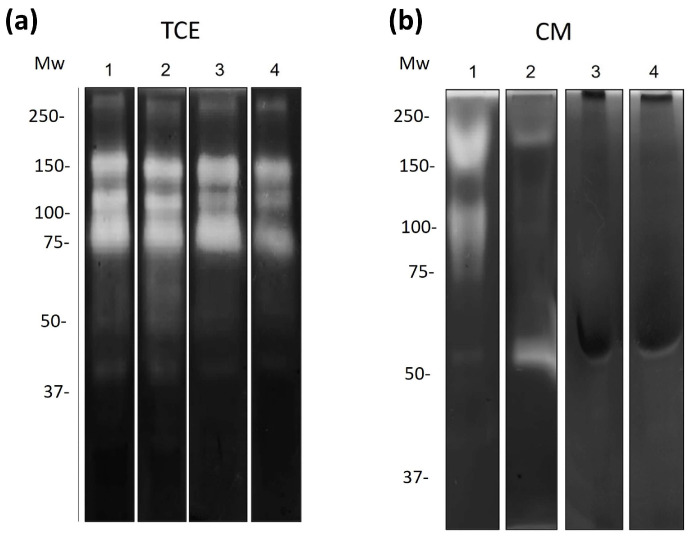
Proteolytic patterns of TCEs and CM from *N. fowleri* after incubation with bLf. (**a**) Zymography in 0.1% porcine gelatin of TCEs from *N. fowleri* obtained after 3, 6, and 12 h of co-incubation with bLf (500 µM) (lanes 2–4). (**b**) Zymography assays of CM after bLf co-incubation at the same times (lanes 2–4). As experimental controls, TCEs and CM from trophozoites without bLf at 12 h were used ((**a**,**b**), lane 1). All the gels were assayed at pH 7.0 at 37 °C. Three independent assays were performed with similar results.

**Figure 4 pathogens-13-00044-f004:**
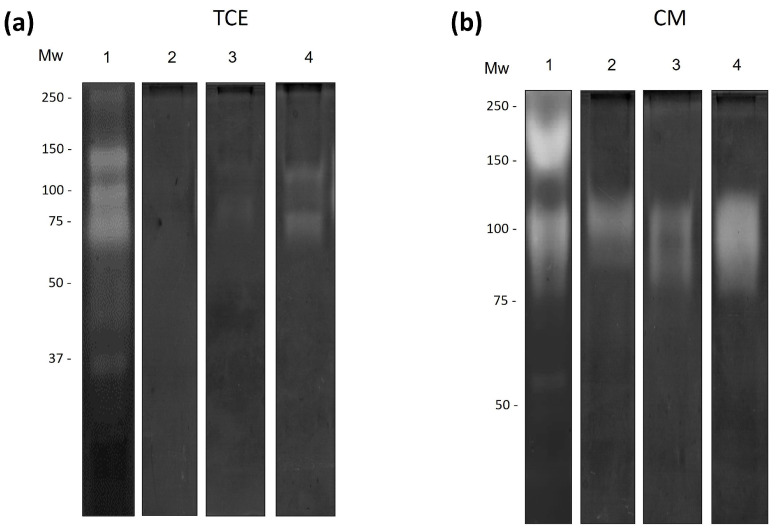
Proteolytic patterns of TCEs and CM from *N. fowleri* after incubation with lysozyme. (**a**) Zymography in 0.1% porcine gelatin of TCEs from *N. fowleri* obtained after 3, 6, and 12 h of co-incubation with cLz (20 µM) (lanes 2–4). (**b**) Zymography assays of CM after cLz (20 µM) co-incubation at the same times (lanes 2–4). As experimental controls TCEs and CM from trophozoites without cLz were used (12 h) ((**a**,**b**), lane 1). All the gels were activated at pH 7.0 at 37 °C. Three independent assays were performed with similar results.

**Figure 5 pathogens-13-00044-f005:**
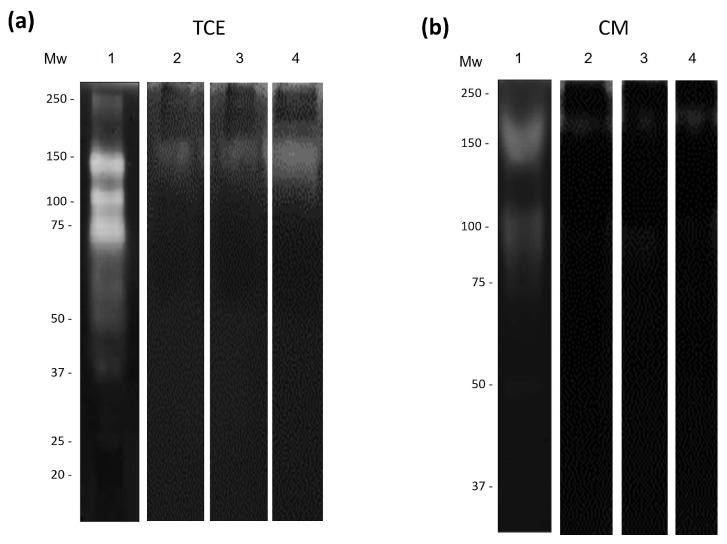
Proteolytic patterns of TCEs and CM from *N. fowleri* after incubation with bLf and cLz. (**a**) Zymography in 0.1% porcine gelatin of TCEs from *N. fowleri* obtained after 3, 6, and 12 h of co-incubation with bLf (500 µM) and cLz (20 µM) ((**a**), lanes 2–4). (**b**) Zymography assays of CM after bLf and cLz co-incubation at the same times (lanes 2–4). As experimental controls, TCEs and CM from trophozoites without antimicrobial proteins at 12 h were used ((**a**,**b**), lane 1). All the gels were activated at pH 7.0 at 37 °C. Three independent assays were performed with similar results.

**Figure 6 pathogens-13-00044-f006:**
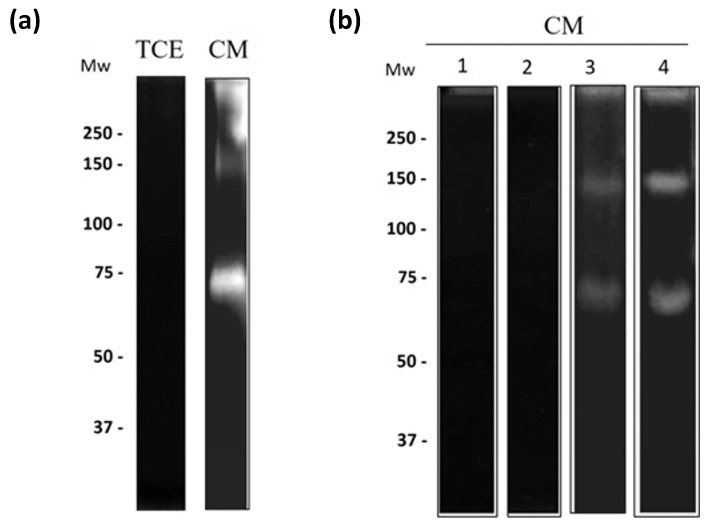
Secreted cysteine proteases degrade bLf (**a**) bLf-Zymogram of TCE and CM from trophozoites pre-incubated with bLf. TCE did not cause protein degradation, while the CM proteases degraded bands at >250, 150, and 73 kDa. (**b**) Partial characterization of proteases from CM in bLf substrate, using protease inhibitors: E-64 (lane 1), pHMB (lane 2), PMSF (lane 3), and aprotinin (lane 4). Gels were activated overnight at 37 °C at pH 7.0. Three independent assays were performed for each experimental condition.

**Figure 7 pathogens-13-00044-f007:**
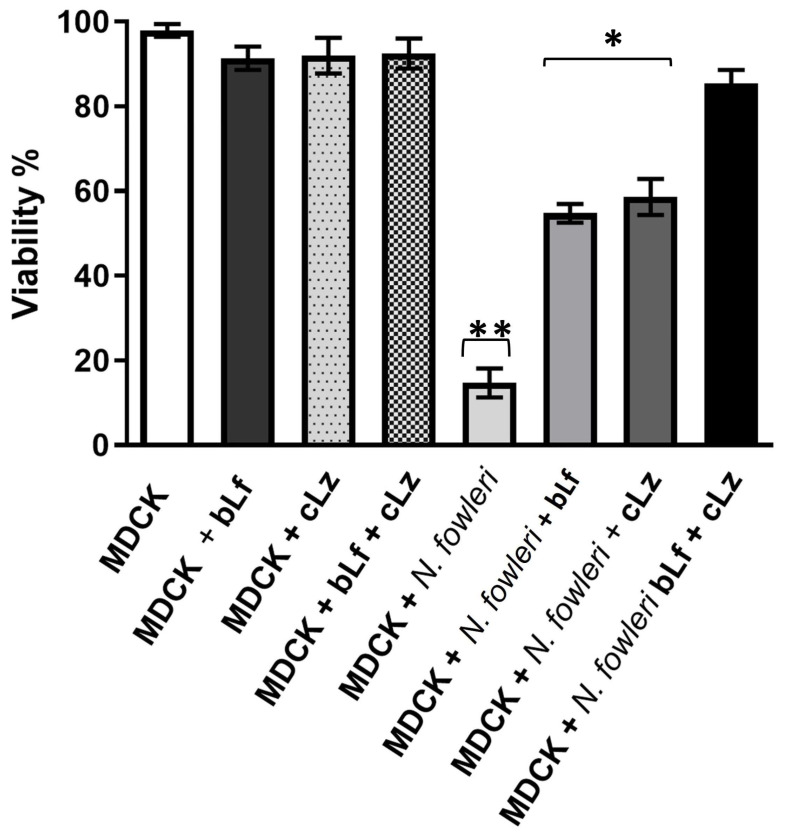
Antimicrobial proteins reduce amoebic virulence in vitro. The cytopathic effect of trophozoites was evaluated on MDCK monolayer with and without trophozoites and 500 µM of bLf and/or 20 µM of cLz or their combination by methylene blue assays. Percentage of the viability of MDCK cells alone (white bar), with bLf (light black bar) and cLz (light gray with dots) or their combination (squares) after 12 h present similar viability, amoebic trophozoites reduce MDCK viability (light gray). Trophozoites with bLf and/or cLz reduce damage (medium gray, deep gray and black). Data analyses were performed with GraphPad Prism 5.0. Bars show the mean ± SD, * *p* < 0.05, ** *p* < 0.01.

## Data Availability

The data presented in this research are available in the manuscript.

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
