# Peer review of "Lactoferrin and Lysozyme Inhibit the Proteolytic Activity and Cytopathic Effect of Naegleria fowleri Enzymes"

_pathogens, 2024, doi:10.3390/pathogens13010044_

Round 1

Reviewer 1 Report

Comments and Suggestions for Authors

Global appreciation:

Naegleria fowleri is a ubiquitous free-living amoeba that causes a rare but one of the most devasting necrotic meningoencephalitis in humans. Infection occurs upon accidental introduction of trophozoite into the nose, after which N. fowleri crosses the cribriform plate to reach the human brain and cause severe destruction of the central nervous system. Nowadays, there is no safe, rapid and effective treatment against this disease. This is mainly due to the limited knowledge on N. fowleri pathogenic mechanisms.

The work presented by the authors aims to study the effect of lactoferrin and lysozyme (natural immune response mediators at the mucosal level) on (i) N. fowleri trophozoites viability and growth and (ii) the amoeba-epithelial cell interaction. The authors show that the proteolytic activities of NF proteases (in particular, cysteine proteases) are differentially impacted by the presence of Lactoferrin (with a more pronounced impact on extracellular proteases) and Lysozyme (which impacts both extra and intracellular proteases). When combined, bLf and cLz have a synergetic effect, with a substantially reduction in the proteolytic activity of enzymes in both NF extracts (TCE and CM), resulting in lower epithelial cell death. The 2 antimicrobials do not induce N. fowleri death or inability to proliferate.  

This work is interesting and of relevance, but I have several questions and comments that I am listing below. I recommend a major revision of the submitted manuscript before acceptance.

 Revision Requests:

1-     Title: Based on the abstract and the results presented by the authors, the authors do not prove that NF proteases allow to escape or evade from the effect of the two antimicrobials but instead that the 2 antimicrobials inhibit the proteolytic activity and cytopathic effect of Naegleria fowleri enzymes. As such, the title should reflect these results.

2-     Abstract

Line 26: please correct to: “The… combination resulted in the inhibition of Naegleria fowleri secreted proteases”.

Line 28: I do not understand the sentence “cysteine participate in its degradation”. Which degradation do the authors refer to? Lactoferrin?

Line 29: please correct to :“ reduced the cytopathic effect of N. fowleri proteases on the MDCK cell line”.

 3-     Introduction

 Lines 81-82: do the authors mean “amoeba survival?”

Line 83: please complete to: “reduced the cytopathic effect due Naegleria fowleri…”

 4-     Materials and Methods

Line 91: please complete “…KUL strain (ATCC 30808) isolated from cerebrospinal fluid.

 Line 100: remove space between b and Lf

Lines 114-115: Please note that MTT assays are designed to check cell viability through the metabolic activity of mitochondria.

Line 120: do you mean MTT instead of MMT? The authors should provide the standard curve as a supplemental data.

Lines 139 and 164: do you mean 0.1% v/v or w/v?

 Line 140: please replace “were incubated or not with” by “were incubated with or without…”

 Line 178: do you mean 1% v/v?

5-     Results

Section 3.1:  The results of NF incubation with different concentrations of bLf and cLz are not presented within the manuscript but they should be at least presented as supplemental data to prove that none of the concentration used are toxic to NF.

The authors selected the maximum concentrations for bLf (500µM) and cLz (20µM). I understand that 20uM cLz is the concentration that can be found physiologically but 500µM bLf is much higher that the values found in the body. Why did the authors preferred to use a high concentration of bLf? The authors should provide an explanation in the discussion section.

Figure 1.

· Please improve the quality of Figure 1a (for instance remove the surrounding square)

· In the YY axis of the Figure 1a, is it really the total number of trophozoites or the concentration (NF/ml)?

·The axis should be within the same range for fig 1b, c and d, ie, with a maximum at 2.0x106.

·Although there is no significant difference between the control NF and NF+peptide within each experiment, there are some differences in terms of growth rate between the 3 experiments. For instance, in Figure 1d, N. fowleri lag phase seems to be longer. Is there an explanation for this? Did the authors calculate the amoeba growth rate in each experiment?

 Section 3.3:  Did the authors check the osmolarity of their culture medium with peptides? I was wondering if hyperosmolarity issues could explain the irregular shape of the trophozoites at 12 and 18 h post inoculation (Figures 2b and d). It is interesting to observes how the trophozoites can rapidly adapt to their new environment.

 Section 3.4.

·  I highly recommend the authors to improve the quality of the Figures 3 and 4

·  While the authors declare in line 249 that there are no differences in protease activity, in Figure 3a, lane 4 is less intense.

· In both Figure 3 and Figure 4, as it is nearly impossible to distinguish the bands in the ladder, please improve the quality or remove it. The authors should always put the tick bars at the right of the number like you did in Figure 3.

·I would suggest writing the hours of incubation (like control, 3,6 and 12h instead of 1.2.3. and 4) at the top of each zymogram.

 Line 265: replace “weakening” by “reduction”.

 Section 3.5: The results provided in this section should be more clearly explained. According to Materials and Methods section, TCE and CM extracts were incubated 1h with protein inhibitors (more or less specific for cysteine proteases) and, afterwards incubated with the Lf-containing zymogram, to see what type of NF proteases would still be able to degrade the antimicrobial protein. Is that correct? 

If NF proteases can degrade Lf, could this explain why trophozoites go back to their “normal” morphology (TEM images)? how could this explain the inhibitory effect of Lf on NF proteases during incubation with MDCK cells? Please provide more details.

In Figure 6, it is impossible to distinguish the bands in the ladder, please improve the quality of the lane or remove it. The authors should always put the tick bars at the right of the number like you did in Figure 3. In Figure 6a and in lane CM bands at the upper level are not clear (they actually do not look like bands). Please improve the quality of the image.

Section 3.6: This section requires substantial rewriting to make it clearer to the reader.

·  The author evaluated the cytopathic effect of Nf trophozoites based on the MDCK cell viability measured by methylene blue. Were the MDCK-containing plates confluent before addition of NF suspension? This is important because 100% viability could be somewhat related to 100% confluence.

· The author should include a sentence to just state that the highest the viability/confluence of MDCK, the lower the cytopathic effect of NF proteases (and the higher the inhibitory effect of peptides).

· Line 310. Please correct to : “Our results showed that the incubation of N.fowleri trophozoites directly onto MDCK cell monolayer led to substantial host cell death, with over 80% decrease in epithelial cell viability….”

· Lines 312-313 – the sentence is not clear. What do the authors means by individual treatment? Were the trophozoites previously incubated with Lf and Lz prior to infection? Or the antimicrobials were incubated with the MDCK cells before infection? This must be also clarified in the Materials and Methods section.

·  Lines 313-314: the authors cannot state that bLf and cLz enhanced MDCK viability but instead they avoided cell death.

· Figure 7. The authors cannot state that antimicrobial proteins reduce amoebic virulence as they did not test it in mice. However, they did reduce the cytopathic effect of NF proteases. In the XX axis, please correct the legends as it is not clear what the authors meant by: MDCK + apo(???) and MDCK +NF+ - bLf? The authors should add the information regarding the control condition MDCK + cLz

·        Lines 322-323: please correct as white bar is for MDCK alone, while black bar is for MDCK + …

 6-     Discussion

 ·   Line 362. Please correct to “the Naegleria fowleri growth curves did not…”

·    Line 376. Please correct “the role of amoebic proteases”…

·  Lines 389-390. Please correct to “The cytopathic effect of NF proteases on epithelial cells were reduced in the presence of Lf and Lz”.

· Line 398-399. Did the authors thought about using a Human Nasal Epithelial Cells Line to perform the studies? Maybe this could be a good starting point before going into in vivo models where sometimes it is difficult to control all experimental parameters.

· Lines 394-395. It is not known what triggers NF proteases to be released into the environment. Maybe the contact with other organisms stimulated this phenomenon. Is it possible that the peptides (namely Lactoferrin that is known to be highly glycosylated) also inhibit NF-MDCK contact hereby decreasing the number (or type?) of amoebae proteases released?

 7-     Conclusion

Lines 408 – 409: do the author mean “allowing a decrease in epithelial cell damage”?

Comments on the Quality of English Language

The quality of the English Language is globally correct but some sentences must be improved.

Author Response

Reviewer 1

Global appreciation:

Naegleria fowleri is a ubiquitous free-living amoeba that causes a rare but one of the most devasting necrotic meningoencephalitis in humans. Infection occurs upon accidental introduction of trophozoite into the nose, after which N. fowleri crosses the cribriform plate to reach the human brain and cause severe destruction of the central nervous system. Nowadays, there is no safe, rapid and effective treatment against this disease. This is mainly due to the limited knowledge on N. fowleri pathogenic mechanisms.

The work presented by the authors aims to study the effect of lactoferrin and lysozyme (natural immune response mediators at the mucosal level) on (i) N. fowleri trophozoites viability and growth and (ii) the amoeba-epithelial cell interaction. The authors show that the proteolytic activities of NF proteases (in particular, cysteine proteases) are differentially impacted by the presence of Lactoferrin (with a more pronounced impact on extracellular proteases) and Lysozyme (which impacts both extra and intracellular proteases). When combined, bLf and cLz have a synergetic effect, with a substantially reduction in the proteolytic activity of enzymes in both NF extracts (TCE and CM), resulting in lower epithelial cell death. The 2 antimicrobials do not induce N. fowleri death or inability to proliferate.  

This work is interesting and of relevance, but I have several questions and comments that I am listing below. I recommend a major revision of the submitted manuscript before acceptance.

Dear reviewer 1. We greatly appreciate the time you have dedicated to reviewing the manuscript, without a doubt, your observations and criticisms are of great help to improve this manuscript. We hope that our answers to his questions are convincing enough. We highlight the changes in the manuscript with yellow color.

Revision requests:

1-     Title: 

Based on the abstract and the results presented by the authors, the authors do not prove that NF proteases allow to escape or evade from the effect of the two antimicrobials but instead that the 2 antimicrobials inhibit the proteolytic activity and cytopathic effect of Naegleria fowleri enzymes. As such, the title should reflect these results.

Answer: Thank you very much for your appreciation, as you have recommended, we changed the title of the manuscript byLactoferrin and lysozyme inhibit the proteolytic activity and cytopathic effect of Naegleria fowleri enzymes”

2-     Abstract

Line 26: please correct to: “The… combination resulted in the inhibition of Naegleria fowleri secreted proteases”.

Answer: Ok, as you have recommended we changed in line 26 to “The bLf and cLz combination resulted in the inhibition of N. fowleri secreted proteases”

Line 28: I do not understand the sentence “cysteine participate in its degradation”. Which degradation do the authors refer to? Lactoferrin?

Answer: You are right, we modified in Line 28 the sentence slightly to imply that cysteine proteases degrade Lactoferrin.

“Additionally, the use of protease inhibitors on bLf-zymograms demonstrated that secreted cysteine proteases participate in the degradation of bLf.”.

Line 29: please correct to :“ reduced the cytopathic effect of N. fowleri proteases on the MDCK cell line”.

Answer: Ok, as you suggest, now we correct in line 29 to “reduced the cytopathic effect of N. fowleri proteases on the MDCK cell line”.

3-     Introduction

 Lines 81-82: do the authors mean “amoeba survival?”

Answer: You are right we mean “amoeba survival in line 81-82

Line 83: please complete to: “reduced the cytopathic effect due Naegleria fowleri…”

Answer: Thank you, as you recommend we complete in Line 83 to “reduced the cytopathic effect due Naegleria fowleri

 4-     Materials and Methods

Line 91: please complete “…KUL strain (ATCC 30808) isolated from cerebrospinal fluid.

Answer: Thank you, you are right, now we complete in line 91 by “KUL strain (ATCC 30808) isolated from cerebrospinal fluid”.

 Line 100: remove space between b and Lf

Answer: Ok, now we remove the space between b and Lf in line 100.

Lines 114-115: Please note that MTT assays are designed to check cell viability through the metabolic activity of mitochondria.

Your observation is correct we include now in lines 114-115 “that mitochondria metabolic activity were evaluated using a 3-(4,5-dimethylthiazol-2-yl)-2,5-diphenyltetrazolium bromide (MTT) assay”

Line 120: do you mean MTT instead of MMT? The authors should provide the standard curve as a supplemental data.

Answer: Thank you, you are right now we change in line 120 “MTT instead MMT”.

Lines 139 and 164: do you mean 0.1% v/v or w/v?

Answer: In lines 139 and 164, we mean 0.1% (w/v). Now we complete in the manuscript.

 Line 140: please replace “were incubated or not with” by “were incubated with or without…”

Answer: Ok, as you recommend, we now replace in line 140 “were incubated or not” by “were incubated with or without”

Line 178: do you mean 1% v/v?

Answer: Yes, in line 178 we mean 1% v/v. Now we complete in the manuscript.

5-     Results

Section 3.1:  The results of NF incubation with different concentrations of bLf and cLz are not presented within the manuscript but they should be at least presented as supplemental data to prove that none of the concentration used are toxic to NF.

Answer: Ok, we added representative pictures of trophozoites incubated with bLf and cLz at 6, 12 and 24 h of co-incubation using the different concentrations evaluated. Moreover, we also included viability graphs of the full assays performed with each concentration and time. 

The authors selected the maximum concentrations for bLf (500µM) and cLz (20µM). I understand that 20uM cLz is the concentration that can be found physiologically but 500µM bLf is much higher that the values found in the body. Why did the authors preferred to use a high concentration of bLf? The authors should provide an explanation in the discussion section.

Answer: Thanks for paying attention to this important point. This decision was made because we did not expect that N. fowleri would be able to survive at concentrations higher than 50 µM of bLf. It has been reported that low concentrations are sufficient to promote the death of other microorganisms, such as E. histolytica, Toxoplasma gondii, and Giardia trophozoites. To demonstrate the significant resistance of N. fowleri, we decided to provide evidence that a concentration ten times higher than the initial proposal of bLf (500 µM) does not cause damage to this microorganism. Furthermore, in Figure 2 of MET we show that incubation with Lfb (500 µM) presents a total morphological recovery without toxicity data.

  • Please improve the quality of Figure 1a (for instance remove the surrounding square)
  • In the YY axis of the Figure 1a, is it really the total number of trophozoites or the concentration (NF/ml)?
  • The axis should be within the same range for fig 1b, c and d, ie, with a maximum at 2.0x106.
  • Although there is no significant difference between the control NF and NF+peptide within each experiment, there are some differences in terms of growth rate between the 3 experiments. For instance, in Figure 1d, N. fowlerilag phase seems to be longer. Is there an explanation for this? Did the authors calculate the amoeba growth rate in each experiment?

Answer:  Ok, we removed the surrounding square in figure 1a.

In the X axis, time is represented, while in the Y axis, the percentage of viability is shown.

The reviewer is correct; we observed these differences that could be explained by the timing of when the experiments were conducted. In fact, bLf experiments were performed in April-June, whereas cLz alone or in combination with bLf were carried out in September-November. This behavior has been observed in our laboratory, but until now, we have not explored the biology behind it. Nevertheless, we calculated the growth rate of amoebas for each experiment, and the data are normalized and compared with the control (amoebas without peptide).

 Section 3.3:  Did the authors check the osmolarity of their culture medium with peptides? I was wondering if hyperosmolarity issues could explain the irregular shape of the trophozoites at 12 and 18 h post inoculation (Figures 2b and d). It is interesting to observes how the trophozoites can rapidly adapt to their new environment.

Answer: No, we did not check the osmolarity of the culture medium with peptides, probably as you think, the osmolarity given by the peptides could be participating in altering the cell shape, and it is a point to test in the future.

 Section 3.4.

  • I highly recommend the authors to improve the quality of the Figures 3 and 4

Answer: Ok, we agree, as you recommend we will improved figures 3 and 4.

  • While the authors declare in line 249 that there are no differences in protease activity, in Figure 3a, lane 4 is less intense.

Answer: You are right, however, the number of the proteolytic bands are the same.

  • In both Figure 3 and Figure 4, as it is nearly impossible to distinguish the bands in the ladder, please improve the quality or remove it. The authors should always put the tick bars at the right of the number like you did in Figure 3.

Answer: Ok, we will remove the bands of the ladder of Figures 3 and 4.

  • I would suggest writing the hours of incubation (like control, 3, 6 and 12h instead of 1.2.3. and 4) at the top of each zymogram.

Answer: Thank you for your comment, in this case we think we describe the incubation hours in detail in the figure below.

Line 265: replace “weakening” by “reduction”.

Answer: As you indicate, we will replace in line 265 “weakening” by “reduction”

 Section 3.5: The results provided in this section should be more clearly explained. According to Materials and Methods section, TCE and CM extracts were incubated 1h with protein inhibitors (more or less specific for cysteine proteases) and, afterwards incubated with the Lf-containing zymogram, to see what type of NF proteases would still be able to degrade the antimicrobial protein. Is that correct? 

If NF proteases can degrade Lf, could this explain why trophozoites go back to their “normal” morphology (TEM images)? how could this explain the inhibitory effect of Lf on NF proteases during incubation with MDCK cells? Please provide more details.

Answer: The objective of this experiment was to know to which type of proteases degrade bLf belong, so, first both TCE and CM were incubated with different types of protease inhibitors for one hour and then the electrophoresis was done in a gel of acrylamide that was copolymerized with bLf as a substrate. The result was that the cysteine proteases of Naegleria fowleri are mainly responsible for the degradation of bLf. On the other hand, we would perform in the future a molecular docking experiments to analyze the interaction between these proteins and proteins of outer membrane from N. fowleri. Its observation is interesting, as mentioned, secreted proteases can degrade bLf, however, when this molecule is degraded it could generate peptides derived from the amino terminal that have been reported to have greater antimicrobial activity (lactoferricins and lactoferrampin) than native bLf. possibly this could participate in this process. In the future, it would be interesting to test these peptides in co-incubation with N. fowleri.

In Figure 6, it is impossible to distinguish the bands in the ladder, please improve the quality of the lane or remove it. The authors should always put the tick bars at the right of the number like you did in Figure 3. In Figure 6a and in lane CM bands at the upper level are not clear (they actually do not look like bands). Please improve the quality of the image.

Answer: You are right, we will remove the bands of the ladder in figure 6 as well as we will put the tick bars at the right of the number in Figure 6.

Section 3.6: This section requires substantial rewriting to make it clearer to the reader.

  • The author evaluated the cytopathic effect of Nf trophozoites based on the MDCK cell viability measured by methylene blue. Were the MDCK-containing plates confluent before addition of NF suspension? This is important because 100% viability could be somewhat related to 100% confluence.

Answer: Thanks for this observation. We have added information to enhance the content of this section as follows: To determine whether N. fowleri trophozoites, when incubated with bLf and cLz, could damage the MDCK cell monolayer, we evaluated the cytopathic effect using methylene blue staining, which was quantified spectrophotometrically. In summary, higher MDCK viability corresponds to a lesser cytopathic effect caused by N. fowleri. You are right, as you say we also take 100% viability as 100% confluence. Therefore we rewrite section 3.6.

  • The author should include a sentence to just state that the highest the viability/confluence of MDCK, the lower the cytopathic effect of NF proteases (and the higher the inhibitory effect of peptides).

Answer: Ok, as you recommend, we will include a sentence that state “the highest the viability/confluence of MDCK, the lower the cytopathic effect of NF proteases (and the higher the inhibitory effect of peptides)”.

  • Line 310.Please correct to : “Our results showed that the incubation of N.fowleri trophozoites directly onto MDCK cell monolayer led to substantial host cell death, with over 80% decrease in epithelial cell viability….”

Answer: In line 310 we will correct to “Our results showed that the incubation of N.fowleri trophozoites directly onto MDCK cell monolayer led to substantial host cell death, with over 80% decrease in epithelial cell viability….”

  • Lines 312-313– the sentence is not clear. What do the authors means by individual treatment? Were the trophozoites previously incubated with Lf and Lz prior to infection? Or the antimicrobials were incubated with the MDCK cells before infection? This must be also clarified in the Materials and Methods section.

Answer: Okay, we used the term "treatment" as a synonym for co-incubation. We changed the word to improve the meaning and avoid misinterpretation..

  • Lines 313-314:the authors cannot state that bLf and cLz enhanced MDCK viability but instead they avoided cell death.

Answer: Thank you for your observation in lines 313-314 we will change the sentence “that bLf and cLz enhanced MDCK viability” by bLf and cLz avoided cell death.

  • Figure 7.The authors cannot state that antimicrobial proteins reduce amoebic virulence as they did not test it in mice. However, they did reduce the cytopathic effect of NF proteases. In the XX axis, please correct the legends as it is not clear what the authors meant by: MDCK + apo(???) and MDCK +NF+ - bLf? The authors should add the information regarding the control condition MDCK + cLz

Answer: Thanks, we have corrected this graph. In fact, we also forgot to include the control for the combination of bLf + cLz co-incubated with MDCK cells.

  • Lines 322-323:please correct as white bar is for MDCK alone, while black bar is for MDCK + …

Answer: You are right, we will correct in lines 322-323 the white bar and the black bar.

 6-     Discussion

  • Line 362. Please correct to “the Naegleria fowleri growth curves did not…”

Answer: As you you recommend we wil correct in line 362 to “the Naegleria fowleri growth curves did not”

  • Line 376.Please correct “the role of amoebic proteases”…

Answer: We will correct in line 376 “the role of amoebic proteases”

  • Lines 389-390.Please correct to “The cytopathic effect of NF proteases on epithelial cells were reduced in the presence of Lf and Lz”.

Answer: As you recommend we will correct to in lines 389-390 to “The cytopathic effect of NF proteases on epithelial cells were reduced in the presence of Lf and Lz”.

  • Line 398-399.Did the authors thought about using a Human Nasal Epithelial Cells Line to perform the studies? Maybe this could be a good starting point before going into in vivo models where sometimes it is difficult to control all experimental parameters.

Answer: You are right your comment is very interesting, in the next experiments we will assay with Human Nasal Epithelial Cell Line before going to in vivo models.

  • Lines 394-395. It is not known what triggers NF proteases to be released into the environment. Maybe the contact with other organisms stimulated this phenomenon. Is it possible that the peptides (namely Lactoferrin that is known to be highly glycosylated) also inhibit NF-MDCK contact hereby decreasing the number (or type?) of amoebae proteases released?

Answer: When we assayed Naegleria fowleri growth only in the culture medium, trophozoites were able to secrete proteases to the medium (Serrano-Luna et al, 2007; Martínez-Castillo et al, 2015). On the other hand, probably it is possible as you say that the derived Lf peptides (Lactoferricin) could be inhibit Naegleria fowleri-MDCK contact and their proteases. It would be interesting to evaluate peptides derived from bLf (lactoferricin or lactoferrampin) in co-incubation with N. fowleri in future trials.

 7-     Conclusion

Lines 408 – 409: do the author mean “allowing a decrease in epithelial cell damage”?

Answer: You are right in lines 408-409 we mean “allowing a decrease in epithelial cell damage”.

Reviewer 2 Report

Comments and Suggestions for Authors

This paper looks at the effect of two proteins (lactoferritin and lysozyme) from the innate immune system on the growth and pathogenicity of Naegleria fowleri a rare but usually fatal infection of the brain.  The paper is well written, and it is very clear. However, there are some shortcomings in the explanation of the rational.

The authors point out that lysozyme is antimicrobial as it cleaves bacteria cell wall components but do not explain why they think that this enzyme should be active against Naegleria. Also, the lactoferritin is postulated to have an effect by sequestering iron but this appears to have been used as an iron-containing protein (ie not apo ferritin) so it would not be expected to have an effect on the amoeba. Naegleria is well known to require iron and that it activates its pathogenicity as is likely to play a role in its ecology (Maciver, S. K., McLaughlin, P. J., Apps, D. K., Piñero, J. E., & Lorenzo-Morales, J. (2021). Opinion: iron, climate change and the ‘Brain Eating Amoeba Naegleria fowleri. Protist. 172, Issue 1, doi.org/10.1016/j.protis.2020.125791). The N. fowleri were grown in a medium containing foetal bovine serum and it this can be exchanged for hemin (Band RN, & Balamuth W (1974) Hemin replaces serum as a growth requirement for Naegleria. Appl Environ Microbiol 28:64–65). Even if the ferritin was partially in the apo form, the amoebae would still obtain iron from the FBS from ferritin?  It is even probable that the amoeba could get enough iron from the lactoferritin itself if required. The other antimicrobial effects of lactoferrin have no expected effects on Naegleria since it is not a bacterium. The Leon-Sicairos, et al, 2006 paper only found activity of these proteins against Entamoeba at very high concentrations and in the presence of IgA. That all being stated, it is of course important to rule out factors as well as identifying those other factors that are likely to prevent infection.

The present title “Proteases of Naegleria fowleri as a mechanism to evade the anti- microbial effect of lactoferrin and lysozyme as natural immune response mediators” is not supported by the work presented. However, the conclusion section seems to be a much better summary (but see below for suggested rearrangement) and also the last sentence of the abstract. The present title infers that lactoferrin and lysozyme have anti-microbial properties against Naegleria fowleri but the data show that they have or such activity against the amoeba.

The iron-bound status of the lactoferritin used in this study should be stated.

Figure 7. The effect of bLf and cLz on trophozoite damage to MDCK cells may be due to the proteins sequestering the activity of the Naegleria proteases. Ideally, a control using some other protein (BSA for example) at the same concentration is necessary to rule out this possibility?  The concentration of bLf and cLz should be stated in the figure legend.

Minor points

Abstract.

 Naegleria fowleri is only ubiquitous in warm countries and so it may be better to delete the word “ubiquitous” here?

The conclusions section may be clearer as

“Our study strongly suggests that bLf and cLz can inhibit secreted N. fowleri proteases causing a decrease in cell damage, but are inefficient in affecting the viability and proliferation of trophozoites.”

Author Response

Reviewer 2

Dear Reviewer 2, we appreciate and thank you for your review of the manuscript. Your observations are very useful for us to improve it; we hope that you find our answers satisfactory. We highlight the requested changes in green

This paper looks at the effect of two proteins (lactoferritin and lysozyme) from the innate immune system on the growth and pathogenicity of Naegleria fowleri a rare but usually fatal infection of the brain.  The paper is well written, and it is very clear. However, there are some shortcomings in the explanation of the rational.

The authors point out that lysozyme is antimicrobial as it cleaves bacteria cell wall components but do not explain why they think that this enzyme should be active against Naegleria. Also, the lactoferritin is postulated to have an effect by sequestering iron but this appears to have been used as an iron-containing protein (ie not apo ferritin) so it would not be expected to have an effect on the amoeba. Naegleria is well known to require iron and that it activates its pathogenicity as is likely to play a role in its ecology (Maciver, S. K., McLaughlin, P. J., Apps, D. K., Piñero, J. E., & Lorenzo-Morales, J. (2021). Opinion: iron, climate change and the ‘Brain Eating Amoeba Naegleria fowleri. Protist. 172, Issue 1, doi.org/10.1016/j.protis.2020.125791). The N. fowleri were grown in a medium containing foetal bovine serum and it this can be exchanged for hemin (Band RN, & Balamuth W (1974) Hemin replaces serum as a growth requirement for NaegleriaAppl Environ Microbiol 28:64–65). Even if the ferritin was partially in the apo form, the amoebae would still obtain iron from the FBS from ferritin?  It is even probable that the amoeba could get enough iron from the lactoferritin itself if required. The other antimicrobial effects of lactoferrin have no expected effects on Naegleria since it is not a bacterium. The Leon-Sicairos, et al, 2006 paper only found activity of these proteins against Entamoeba at very high concentrations and in the presence of IgA. That all being stated, it is of course important to rule out factors as well as identifying those other factors that are likely to prevent infection.

Answer: You are right in your observation about how lysozyme functions against bacteria. However, in our group, we reported in 2006 (Leon-Sicairos et al, 2006) that Lysozyme has an amoebicidal effect, alone or in combination with lactoferrin and IgA. In this manuscript, we suggested that Lysozyme could bind to lipopeptidophosphoglycan described previously in Entamoeba histolytica (Isibasi et al, 1982). Based on this data we assayed the effect of Lysozyme in trophozoites of N. fowleri. On the other hand, in the experiments we used bLf partially charged with iron (5 mg/100 g). This concentration of bLf is able to kill bacteria. In other assays with holo-Lf (lactoferrin with two atoms of iron), we observed a good growth of E. histolytica (León- Sicairos et al, 2005) and in Acanthamoeba genotype T4 (not reported data). FBS contains manly apo-Tf. Ferritin is mainly in the liver; this protein is a storage protein. Lactoferrin is mainly in milk and in mucosal secretions.

 The present title “Proteases of Naegleria fowleri as a mechanism to evade the anti- microbial effect of lactoferrin and lysozyme as natural immune response mediators” is not supported by the work presented. However, the conclusion section seems to be a much better summary (but see below for suggested rearrangement) and also the last sentence of the abstract. The present title infers that lactoferrin and lysozyme have anti-microbial properties against Naegleria fowleri but the data show that they have or such activity against the amoeba.

Answer: You are right. As reviewer 1 suggested we changed the title by Lactoferrin and lysozyme inhibit the proteolytic activitiy and cytopathic effect of Naegleria fowleri enzymes” that represents better the work done.

The iron-bound status of the lactoferritin used in this study should be stated.

Answer: Thank you for your observation, the bLf contains (5mg of iron /100g). We will state in the manuscript.

Figure 7. The effect of bLf and cLz on trophozoite damage to MDCK cells may be due to the proteins sequestering the activity of the Naegleria proteases. Ideally, a control using some other protein (BSA for example) at the same concentration is necessary to rule out this possibility?  The concentration of bLf and cLz should be stated in the figure legend.

Answer: You are right in your observation, however, when we used MDCK cells incubated with bLf or cLz we do not see any damage, MDCK culture medium includes FBS that contains different proteins like albumin. Nevertheless, as you said in the future we need to add different harmless proteins. We add the requested information in the figure legend.

Minor points

Abstract.

 Naegleria fowleri is only ubiquitous in warm countries and so it may be better to delete the word “ubiquitous” here?

Answer: You are right; we will delete the word ubiquitous in the manuscript.

The conclusions section may be clearer as

“Our study strongly suggests that bLf and cLz can inhibit secreted N. fowleri proteases causing a decrease in cell damage, but are inefficient in affecting the viability and proliferation of trophozoites.”

Answer: We agree with you, we will change the conclusion to “Our study strongly suggests that bLf and cLz can inhibit secreted N. fowleri proteases causing a decrease in epithelial cell damage, but are inefficient in affecting the viability and proliferation of trophozoites.”

Reviewer 3 Report

Comments and Suggestions for Authors

Dear Authors,

Please see an attached file.

Comments on the Quality of English Language

I believe that extensive editing of English language required.

Author Response

Reviewer 3

The manuscript entitled “Proteases of Naegleria fowleri as a mechanism to evade the antimicrobial effect of lactoferrin and lysozyme as natural immune response mediators” showed that bovine milk lactoferrin (bLf) and chicken egg lysozyme (cLz) could inhibit secreted proteases of Naegleria fowleri, and thereby, could reduce cytopathic effects of the parasite without affecting the viability and proliferation of the trophozoites. The results were clearly shown, but extensive English editing may be required and some controls are missing.

[Major concern]

The authors suggested that bLf and cLz inhibited N. fowleri secreted proteases, but the reviewer is not sure whether bLf and cLz actively inhibited the proteases. For example, we generally use trypsin to detach adherent cells as authors may be aware. Before treating the cells with trypsin solution, we need to wash the culture with PBS because, otherwise, trypsin’s protease activities will be consumed for digesting abundant serum proteins in culture media and not efficiently be utilized for detaching the cells. I wonder whether this is also the case in the present studies. Abundant bLf or/and cLz may exhaust the proteases non-specifically. Authors need to include the results with non-antimicrobial proteins (eg. bovine serum albumin) to show that bLf and cLz specifically inhibited N. fowleri secreted proteases in Figure 3 or 7. “MDCK+cLz” (control for “MDCK+N. fowleri+cLz”) is also missing in Figure 7.

Dear reviewer 3, we greatly appreciate the time dedicated to reviewing the manuscript, which will be very useful to have a better version of it, we hope that our answers satisfy your questions. We highlight the requested corrections in color blue.

Before sending the manuscript to the magazine, we did the editing in English with Editage Company. If you wish, we can forward the manuscript to this company again for review in English.

Regarding the result of the inhibition of proteolytic activity by bLf and cLz, we have observed the same inhibition activity using lower concentrations of bLf with the bacteria  Manhemia hemolytica (Ramírez-Rico et al, 2021) and Actinobacillus pleuropneumoniae (Luna-Castro et al 2014) with the protozoan Entamoeba histolytica (Data not shown), so we do not think that it is the excess of these proteins causes the inhibitory effect of the proteolytic activity of N. fowleri. We observed the same thing in a recent article that we published with Acanthamoeba genotype T4 (Ramírez-Rico et al, 2023). Furthermore, in our experiments we did not use trypsin to detached culture cell. Once the MDCK cells reach 100% confluence, we carry out the incubation with Naegleria fowleri alone or with the antimicrobial proteins and measure the viability of these cells. On the other hand, by zymography we observed the inhibition of proteolytic activities prior to an incubation of bLf or cLz with N. fowleri. On the other hand, in figure 7, we now include the control of MDCK co-incubated with cLz.

 [Minor concerns]

  1. (line 100) “b Lf” would be “bLf”. 2. (line 164) “using as substrate bLf at 0.1%” might be “using 0.1% bLf as substrate”. 3. (lines 265 and 266) “Figure 4b” would be “Figure 4a”. 4. (line 267) “lanes 4 and 5”, there was not lane “5”. 5. (line 270) “lanes 2-5” would be “lanes 2-4”. 6. (Figures 4, 5 and 6) Molecular sizes need to be “250-, 100-, etc.” but not “-250, -100, etc.” 7. (Figure legend of Fig.5): please revise carefully. a. (line 282 and 283) “TCEs” would be “TCE” b. (line 282) “Lz” would be “cLz” c. (line 284) “lanes 2-5” would be “lanes 2-4” d. (line 285) “(b, lanes 2-4)” would be “(lanes 2-4) e. (line 285-286) “TCE and CM from trophozoites without antimicrobial proteins at 12 h were used” need to be reconsidered. 8. (lines 290 and 292) There were not “lane 1” and “lane 2” in Figure 6a. 9. (lines 294 and 296) “lanes 1-2” and “lanes 3-4” would be “lanes 1 and 2” and “lanes 3 and 4”, respectively. 10. (line 295) “non-specific cysteine protease inhibitors” would be “serine protease inhibitors” as in line 167. 11. (line 296) “Zymograms of cLz were obtained; however” would be “Zymograms of cLz were not obtained; because”. 12. (Figure legend of Fig.7) Annotations of the bars need to be corrected. 13. (line 346) “18 uM” should be “12.5 uM”. 14. (lines 389-390) “The cytopathic effects of N. fowleri on epithelial cells were reduced” by which treatment? 15. (lines 394-395) “Probably, bLf and cLz can bind directly to proteases, consequently reducing their activity.” would not reflect the results in “3.5 Cysteine proteases from N. fowleri degrade bLf”. “their activity” would be “their activities”.
  2. (line 100) “b Lf” would be “bLf”.

Answer: You are right; we changed “b Lf by “bLf” in line 100

  1. (line 164) “using as substrate bLf at 0.1%” might be “using 0.1% bLf as substrate”.

Answer: You are right; we changed  “bLf at 0.1%” by “using 0.1% bLf as substrate” in line 164

  1. (lines 265 and 266) “Figure 4b” would be “Figure 4a”.

Answer: You are right; we changed “Figure 4b” by “Figure 4a” in lines 265 and 266

  1. (line 267) “lanes 4 and 5”, there was not lane “5”.

Answer: You are right; we changed “lanes 4 and 5 by lanes 3 and 4 in line 267

  1. 5. (line 270) “lanes 2-5” would be “lanes 2-4”.

Answer: You are right; we changed “lanes 4 and 5” by “lanes 2-4” in line 270

  1. (Figures 4, 5 and 6) Molecular sizes need to be “250-, 100-, etc.” but not “-250, -100, etc.”

Answer: Thank you for your comment; we changed the figures 4, 5 and 6.

  1. (Figure legend of Fig.5): please revise carefully. a. (line 282 and 283) “TCEs” would be “TCE” b. (line 282) “Lz” would be “cLz” c. (line 284) “lanes 2-5” would be “lanes 2-4” d. (line 285) “(b, lanes 2-4)” would be “(lanes 2-4) e. (line 285-286) “TCE and CM from trophozoites without antimicrobial proteins at 12 h were used” need to be reconsidered.

Answer: Ok, we agree with your observation, we changed all your observations.

  1. (lines 290 and 292) There were not “lane 1” and “lane 2” in Figure 6a.

Answer: As a suggestion from reviewer 1, lane 1 of Figure 6a which corresponded to the molecular weight markers, was eliminated, leaving the proteolytic activities of TCE at 1 and the proteolytic activities of CM at 2 in lines 290 and 292.

  1. (lines 294 and 296) “lanes 1-2” and “lanes 3-4” would be “lanes 1 and 2” and “lanes 3 and 4”, respectively.

Answer: You are right “lanes 1-2 change to lanes 1 and 2 in line 294 and “lanes 3-4 change to lane 3 and 4 in line296.

  1. (line 295) “non-specific cysteine protease inhibitors” would be “serine protease inhibitors” as in line 167.

Answer: You are right; we changed “non-specific cysteine protease inhibitors” by “serine protease inhibitors” in line 295 as in line 167.

  1. (line 296) “Zymograms of cLz were obtained; however” would be “Zymograms of cLz were not obtained; because”.

Answer: You are right; we changed “Zymograms of cLz were obtained by). Zymograms of cLz were not obtained; because copolymerization of acrylamide with this antimicrobial protein was not successful (data not shown) (lines 296-298).

  1. (Figure legend of Fig.7) Annotations of the bars need to be corrected.

Answer: You are right; we corrected annotations of the bars.

  1. (line 346) “18 uM” should be “12.5 uM”.

Answer: You are right 18 µM changed to “12.5 µM in line 347.

  1. (lines 389-390) “The cytopathic effects of N. fowleri on epithelial cells were reduced” by which treatment?

Answer: You are right; We complete the sentence by “The cytopathic effect of N. fowleri proteases on epithelial cells were reduced in the presence of bLf and cLz” in lines 389-390.

  1. (lines 394-395) “Probably, bLf and cLz can bind directly to proteases, consequently reducing their activity.” would not reflect the results in “3.5 Cysteine proteases from N. fowleri degrade bLf”. “their activity” would be “their activities”.

Answer: The results of assay 3.5 by zymography using bLf as a substrate, the trophozoites did not have a prior incubation with bLf or cLz so the N. fowleri proteases have not been affected by these antimicrobial proteins. On the other hand, when we previously incubated with bLf or cLz, these proteins could probably interact with the proteases or with the trophozoite membrane and alter its secretion mechanism and perhaps for these reasons the protelytic activity was inhibited. In addition, peptides derived from Lf (Lactoferricins, lactoferrampins) can be generated that have been reported to have greater antimicrobial activity than native Lf. However, in future studies we could perform different assays such as molecular docking with bLf or Lz with the outer membrane proteins of N. fowleri to test these hypotheses.

Round 2

Reviewer 1 Report

Comments and Suggestions for Authors

Dear Authors,
Thank you for taking my comments and suggestion into consideration to improve your manuscript. I have no additional comments.

Reviewer 3 Report

Comments and Suggestions for Authors

Dear Authors,

Thank you very much for responding to all of my concerns.
I will highly recommend the manuscript for publication in the journal.

Best regards,